# Walking in Each Other's Footsteps: Do Animal Trail Makers Confer Resilience against Trampling Tourists?

**David B. Croft**

School of Biological Earth & Environmental Sciences, UNSW Sydney, NSW 2052, Australia; d.croft@unsw.edu.au

**Abstract:** Modern humans, and other hominins before them, have walked across the landscapes of most continents for many millennia. They shared these landscapes with other large animals, especially mammalian herbivores and their predators, whose footsteps defined trails through the vegetation. Most of the diversity in the wild species is now concentrated in protected areas and visited by large numbers of tourists who may walk amongst them. This review examines the literature about medium-large animal and tourist trampling impacts to uncover any marriage between animal ecology and nature-based tourism research. Methodology is comparable. Animal ecology has focused on the propagation of grazing and trampling effects from a point source (usually water). Tourism research has focused on trail structure (formal/informal, hardened, wide/narrow) and the propagation of effects (especially weeds) into the hinterland and along the trail. There is little research to substantiate an evolutionary view of trampling impacts. At least tourists venturing off formed trails may reduce impacts by following animal trails with caveats, such as risk of encounters with dangerous animals and disruption of animal behavior. This is an under-studied topic but a fertile ground for research, aided by modern tools like trail cameras and geographically enabled devices borne by tourists.

**Keywords:** trampling; nature-based tourism; animal ecology; animal trails; tourist trails; wildlife tourism

---

## 1. Introduction

In the human evolutionary tree, Australopithecines were likely the first hominins to walk bipedally on African landscapes around 3.5 million years [1]. Modern humans, *Homo sapiens*, walked out of Africa as early as 220,000 years ago [2] or at least within a range from 126–74 thousand years ago [3]. These early humans collectively walked very long distances reaching Australia (Sahul) 50–65 thousand years ago [4]. In Australia, in the absence of 'beasts of burden' [5], they continued to exclusively walk across the landscape until European settlement, in 1788, brought new species and transport technologies. Thus humans have been walking on the vegetated substrate of most continents for many millennia. The impact of such activity has been formally studied in ecology since seminal works like Bates' 1935 study of the vegetation of footpaths and other passageways [6]. Subsequently, trampling of vegetation has been at the forefront of ecological impacts identified in nature-based tourism and recreation (e.g., [7]).

Even so, in these natural settings other species, such as mammalian herbivores, are trampling vegetation and leaving well-defined paths. In animal ecology, these impacts have been studied in the context of sustainable livestock production and the maintenance of a functional landscape, alien and invasive species' management, and wildlife management in protected areas. Impacts are most visible in semi-arid and arid rangelands and most intense around point sources of water creating a region of heavy impact known as the piosphere [8]. Such effects of other species led Ejrnaes [9] to propose that 'the elephant in the room' when discussing human trampling impacts is the elephant. In other words, we should take an evolutionary approach to such effects in recognition of past long-term impacts, including those of high magnitude from contemporary or extinct megafauna. A key difference is,

of course, that human tourists typically do not crop the vegetation they amble across. Thus mammalian herbivores, whether livestock or wildlife, leave footprints and grazing impacts.

The aim of this short review is to examine (1) whether animal ecology can inform the appropriate management of trampling impacts by nature-based tourists, (2) the degree to which paths of trampled vegetation can be shared between people and wildlife so that the impacts are not separate and additive, and (3) perceptions of landscape degradation and the relationship between trampling resistance and the medium-large mammal fauna. The review focuses on terrestrial landscapes but recognizes a significant literature about trampling on reefs, saltmarshes, and tidal zones.

## 2. Trampling Impacts of Hooves and Hard Landings

In the modern world, we have displaced a diverse guild of mammalian herbivores and their attendant predators and left them to aggregate on reserves and protected areas. In their place are settlements, agriculture and pastoralism. The latter aggregates one or a few species of domesticated mammalian herbivores at high densities in livestock production systems. Tourists seeking nature-based experiences like wildlife viewing are likewise aggregated in a few wild places. Thus former free-roaming populations of wild animals and people may have trod lightly on the landscape but are now aggregating at high densities even if the individual people (nature-based tourists) are transitory. The consequence inevitably is that the weight of the footfalls is heavier, and the resilience of vegetation and soils are acutely tested.

The piosphere epitomizes the trampling impacts of aggregations of livestock or wildlife at high densities (Figure 1a) [10]. In production systems, like sheep grazing in the Australian rangelands, the piosphere is a zone of hardened and denuded soil with scant vegetation that is unpalatable to livestock [11,12] (Figure 1b). Washington-Allen and colleagues [13] expanded the definition to include "any concentrated animal or anthropogenic impact that radiates from an area of concentration". They created a remote-sensing tool to detect piospheres and tested this in a sagebrush site in Utah USA. Their definition and tool can be applied to both animal and human trampling. However, most studies are on animals, especially livestock, from areas as diverse as the Chaco dry forests of Argentina [14], the rangelands of Iran [15] and the steppe grasslands of Mongolia [16].

The intensity of the piosphere is a function of the sociality of the species using it, not just the number of individuals visiting during a day. The herding nature of domesticated herbivores and of many wild species like medium-large antelope concentrates the impact of many hooves at a single point in time. Some species, like kangaroos in the Australian rangelands do not show water-focused grazing [17,18] and visit water sources as singletons or small loose aggregations (mobs) so that, in the absence of livestock, the piosphere diminishes. There is also some evidence that this is because they have 'soft' feet causing less disruption of the soil surface than hard hooves [19–21].

The piosphere provides several insights that may be useful to manage environmental impacts of 'trampling' tourists. Firstly, there may be a density effect. Singletons, pairs and small groups traversing a landscape at staggered intervals (like kangaroos) may leave less impact than a large group (e.g., a busload) traversing a landscape together (like sheep) even if the total foot traffic is the same in a day. Secondly, context may modify the perception of a trampling effect. In Figure 1a, the concentration and diversity of species on view may override any perception of landscape degradation. Whereas Figure 1b may lead to perceptions of over-stocking, unsustainable practices, desertification and eventual impoverishment of people and landscape. Thirdly, there are significant trampling effects in which the nature-based tourist does not directly participate. In the context of Figure 1a, the tourist is confined to a common viewing access and a vehicle and does not walk on this landscape due to dangerous predators, potentially aggressive and harmful species, and unwarranted disturbance of the fauna they have come to view. In Figure 1b, the tourist visits the village for a cultural experience but does not tend the livestock.

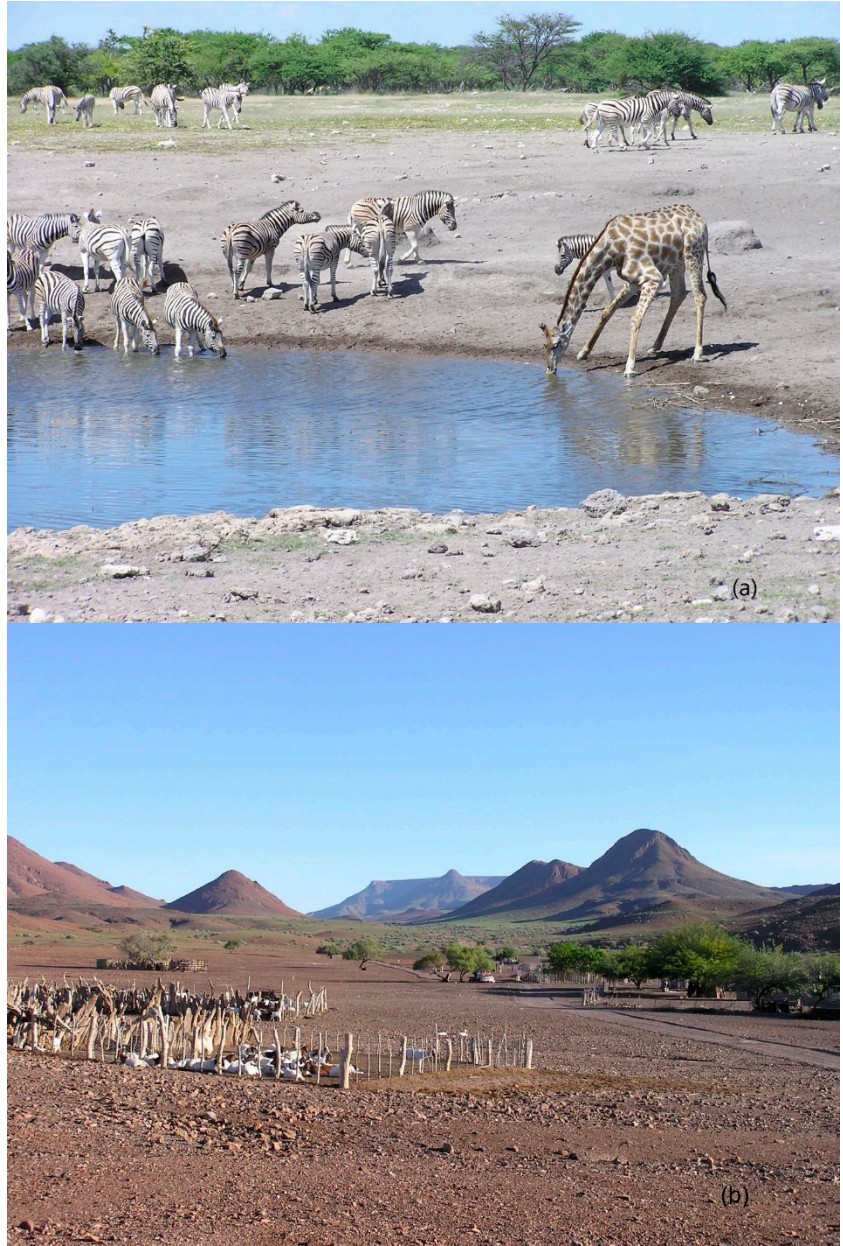

**Figure 1.** (**a**) Example of a piosphere created by trampling and grazing in Etosha National Park, Namibia. (**b**) Example of heavy trampling and grazing impact of livestock leading to denuded vegetation in Damaraland, Namibia (Images: D.B. Croft).

The piosphere nonetheless creates a quandary for the land manager, especially in protected areas, where fauna need to be balanced with flora and the functionality of landscapes needs to be maintained (i.e., not irreversibly degraded) [22]. Where fauna are the attraction, as in many African national parks, tourists can be corralled to their own vehicles or tourist vehicles and delivered by a road network to viewing points or platforms (Figure 2a). Thereby the tourists' impact is confined to the road network. However, aggregating the fauna at point sources like artificial watering points (Figure 2b) creates high grazing and trampling impacts and can have unexpected consequences in the conservation of both fauna and flora.

For example, a program to introduce more watering points to the northern section of Kruger National Park in South Africa included a goal of conserving rare Roan Antelope (*Hippotragus equinus*) as well as distributing more visitation to this under-utilized section of the park [23]. One consequence

was actually a decline in the Roan Antelope, which was not attributed to competition for forage from previously transitory herbivores that took up residence, but rather predation from the lion prides that accompanied them. The program was subsequently truncated and many artificial watering points were closed [24]. Even so not all herbivore species in this Park are drawn to artificial waterholes. Grazers aggregate around them but browsers and mixed feeders associate with natural water sources like rivers [25]. In Hwange National Park in Zimbabwe, Chamaille'-Jammes and colleagues [26] found a weak expression of a piosphere in terms of loss of the heterogeneity of woody vegetation in a landscape with a large African Elephant (*Loxodonta africana*) population. They attributed this in part to the diversity of herbivorous mammals. In contrast, Landman and colleagues [27] studied long-term (31-year) effects of elephant induced piospheres in Addo Elephant National Park in South Africa. The effects were complex and usually manifested as a zone of grassland that replaced shrubland in about a 300-m radius around a watering point. They advocated less water provisioning to avoid degradation of important succulent thicket habitats. They recognized the consequence would be more intense impacts at residual water points.

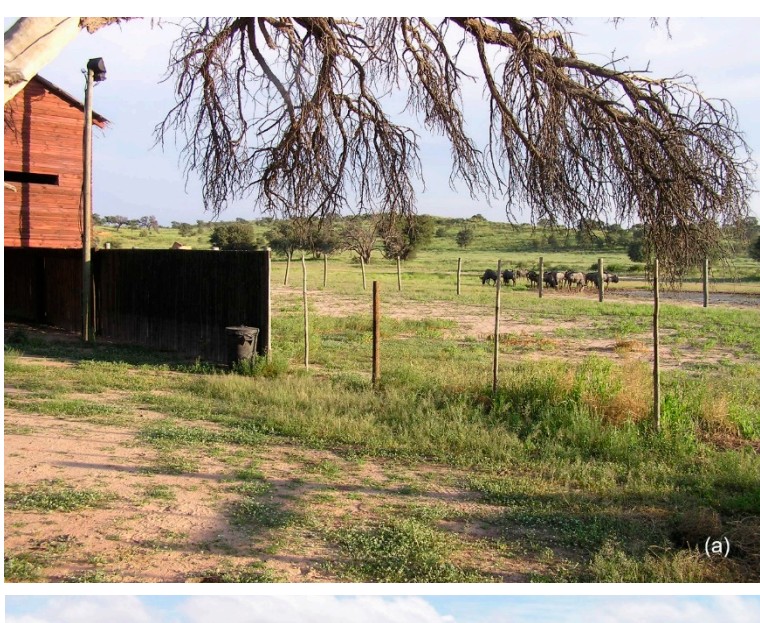

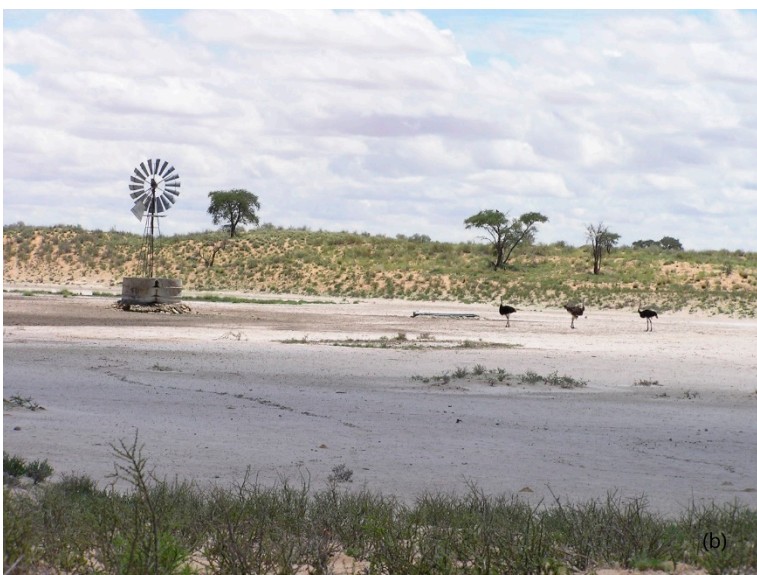

**Figure 2.** (**a**) Hide with night illumination overlooking a watering point in the Kgalagadi Transfrontier Park in South Africa. (**b**) Bore supplying water to fauna in Kgalagadi Transfrontier Park (Images: D.B. Croft).

In Australia, this quandary is approached from the opposite direction. Many National Parks in the rangelands have been formed from abandoned or acquired pastoral leases and come with pastoral infrastructure such as artificial watering points (bores and earthen tanks) [28] and former holding paddocks around livestock management points such as woolsheds and stockyards. Closure of artificial watering points has been the norm unless there is some utility such as a water supply to residents/visitors in the Park (e.g., [29]). In that case they are fenced off to mammalian herbivores whether native or feral. The arguments for closure are typically based on artificiality at odds with a natural ecosystem and/or utility to manage grazing impacts of feral and introduced herbivores like goats [18]. These water sources impose a cost as they require maintenance (de-silting and repair of banks when dry) or infrastructure such as pumps, holding tanks and troughs. Thus economics typically overrides the ecological context. Pastoralism brought massive and often degradative changes to the Australian rangelands and this included loss of natural water holes through sediment infilling [30]. Thus artificial watering points are not entirely additive to water holdings in the landscape but may be substitutional. Indigenous cultures maintained water holdings in arid Australia [31] and so anthropogenic intervention has a long history. Utility for visitor safety in water-deficient environments, improved visitor satisfaction with wildlife viewing platforms [32] and support for water-dependent species [33] should have due consideration. If degradative impacts are not persistent and recovery is occurring, albeit slowly [17], then a rational approach should pertain to removing water holding structures from drought-affected landscapes. Pastoralism in the Australian rangelands has driven them to a new state [34]. Removal of livestock will not return the rangelands to a pre-pastoral state especially with a future of rapid climate change. If degradative processes at point water sources (natural or man-made) are at bay then new utility can be sought in the absence of pastoralism and the African experience (with caveats) supports tourism infrastructure as a valid one.

These examples are drawn from very large national parks in the rangelands of Southern Africa and Australia that share climate drivers in the Southern Hemisphere. Even so the piosphere is likely to exercise the management of environmental impacts wherever medium-large mammals are a tourism drawcard and water (or other essential nutrients) are concentrated at point sources. In a tourist destination, a small carpark with radiating informal hiking trails extending from it may share degradative characteristics with the piosphere, Thus there may be synergies in analyzing and ameliorating environmental impacts for both human and animal foot traffic.

## 3. Tourists and Wildlife in Each Other's Footsteps

If landscapes are already dissected by pathways created by medium, large or mega mammals then the trampling impact of ambulatory tourists may be lessened by walking in their footsteps. There are, of course, caveats to this approach which are discussed in the following. Such pathways are most obvious in an open landscape like the arid rangelands and tundra. For example, dry lichen mats are particularly sensitive to trampling by Reindeer (*Rangifer tarandus*) [35]. From this experimental research, one can postulate that each Reindeer can leave trails up to 5 km in their daily movements across tundra with extensive mats of dry lichen. Such trails have not been mapped and there is a paucity of information on animal trail networks. There has been some research in testing foraging models against field data on foraging trail networks (e.g., hippopotamus [36]) but most current efforts relate to placement of camera traps on animal trails to monitor trail users (e.g., mammal communities in Ruaha National Park Tanzania [37]).

I and two assistants mapped the pathways (animal trails) left by kangaroos (predominantly the Red Kangaroo *Osphranter rufus*, with smaller contributions from the Common Wallaroo *Osphranter robustus* and Eastern Grey Kangaroo *Macropus giganteus*) radiating from three water points in Sturt National Park, New South Wales, Australia. The mapping was conducted using a hand-held GPS receiver (Garmin devices) by starting at the water source (a large earthen tank from the pastoral era) and walking along the path until it became ill-defined. If the path bifurcated, then each branch was mapped by back-tracking and walking it. The end-points were typically a shelter belt which, in the

stony downs landscape, were sparse and often lineal along an intermittently flowing water course. The resultant pathways are shown for one water point, North Torrens Tank (Figure 3). Pathways radiate from the water point but there is an extensive network beyond its immediate zone of influence. Discontinuities arose where pathways crossed rocky outcrops and became indistinct.

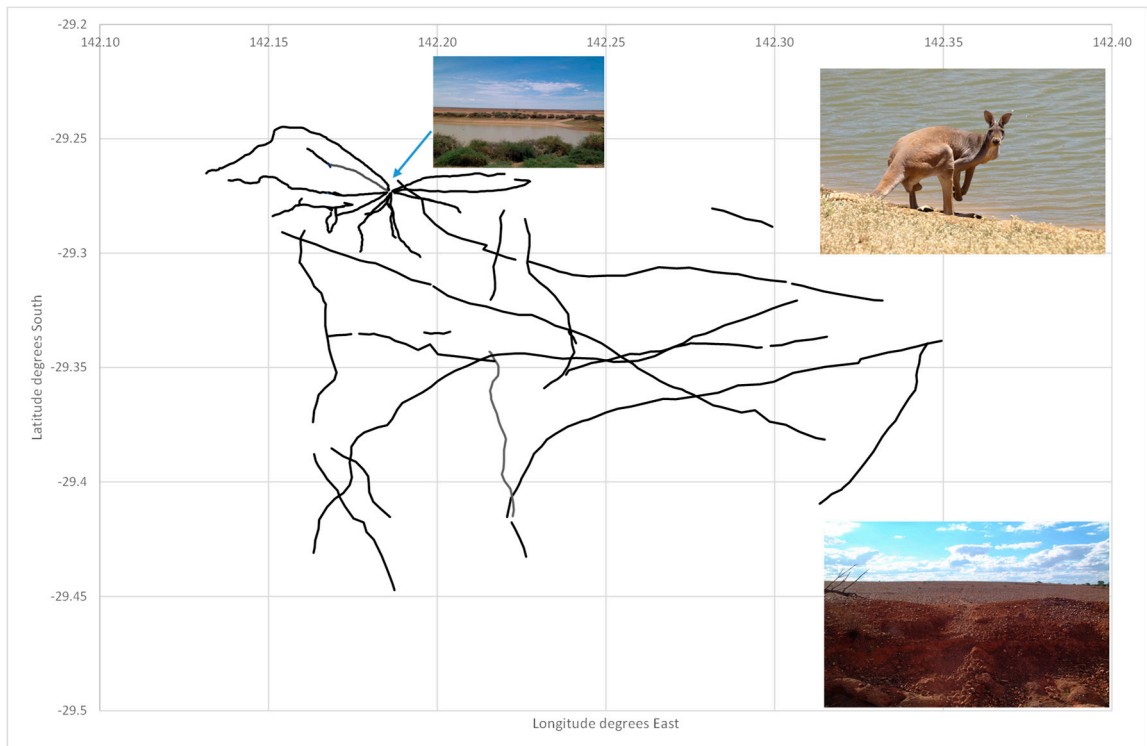

**Figure 3.** Map of pathways (example in lower right) created predominantly by red kangaroos (pictured in upper right) radiating from and beyond an earthen tank (pictured in upper left) on the stony downs of Sturt National Park, Australia (Images: D.B. Croft).

Pathways of utility to the ambulatory tourist are found in all landscapes with terrestrial animals of commensurate size (or bigger) to humans, when those animals traverse their habitat in a regular pattern. They may not always be distinct. In tropical savannas they may be overgrown by tall grasses in the wet season (Figure 4a) and covered in a litter of dry grass and leaves in the dry season (Figure 4b). Even so they will likely provide the easiest passage through dense vegetation. Likewise they may provide the least-effort traverse across hilly terrain in relation to slope angle as studies on wildlife [38] and livestock [39] in the USA have shown. Their contribution to vegetation biomass loss in landscapes may be small. For example, Plumptre [40] determined 1.5% of vegetation in Parc National des Volcans, Rwanda, showed permanent trampling damage from Mountain Gorilla (*Gorilla gorilla beingei*), African Buffalo (*Syncercus caffer)* and African Elephant with about 0.01% trampled daily.

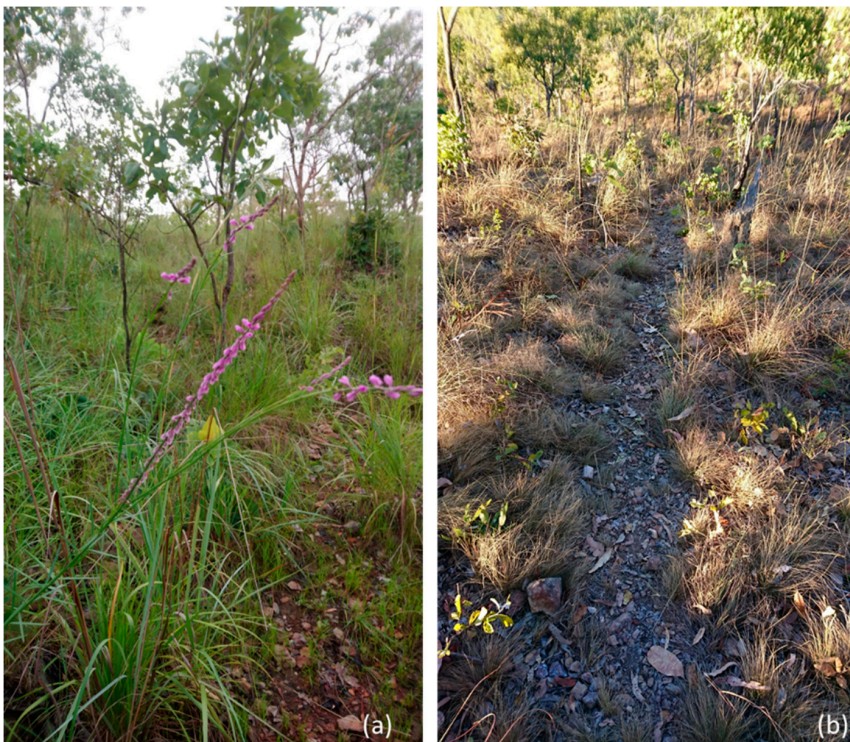

**Figure 4.** Pathway formed by regular progress of wallaroos (*Osphranter antilopinus*, *O. robustus*) along a ridge top in a tropical savanna in (**a**) wet and (**b**) dry seasons (Images: D.B. Croft).

In conservation reserves, tourists are typically confined to formed trails that progress between points of interest. Even so users may go off trail and the result may be extensive damage to vegetation in a landscape [41]. For example, in the Danube Floodplains National Park in Austria 61% of hikers used designated trails, 21% used non-marked paths and 18% went off-trail [42]. There may be penalties for going off-trail but policing may be scant. In large, remote reserves such as found in outback Australia, or Africa (e.g., Namibia, Botswana) tourists gain access using a road network and may be directed to vantage points with parking or points of interest along short walking trails. Usage by visitors driving typically greatly exceeds that of hikers [43]. Driving off-road is strongly discouraged and may attract heavy penalties if policed. The constraints on going off-trail by hikers are largely self-imposed (e.g., fear of getting lost or encountering a dangerous animal). Direction of hikers to animal pathways would minimize additive trampling impacts. It will orient them to sites where animals of interest may aggregate, and to useful landscape features such as water and shelter. Progress along such pathways is likely to be of least effort. A pathway may also become the foundation of a visitor experience with the addition of markers and signage, and allocation to a formal trail network. Even so use of animal pathways comes with a number of caveats.

(1)　Predators such as *Panthera* spp. (Lion, Tiger, Jaguar and Leopard) may monitor animal pathways in order to ambush prey. Attacks on people by these large cats cause hundreds of deaths annually [44]. Most risk is at night, especially to sleeping individuals (e.g., [45]). Management of the activity periods of tourists can minimise risk along with the knowledge of a skilled, and possibly armed, guide. In my personal experience, visitors to the Okavango Delta in Botswana are led on daytime walking tours amongst potentially dangerous animals with a skilled guide with minimal adverse consequences.

(2)　Danger may reside not on the pathway but at its endpoints. For example, pathways to water in Northern Australia could lead a tourist into crocodile (*Crocodylus porosus*) habitat and, if they enter the water, there is a high risk of fatal attack [46]. Similarly, in Africa pathways in the vicinity of water made by night-grazing Hippopotamus (*Hippopotamus amphibious*) may lead to an encounter with

this aggressive and dangerous species [47]. Again, risk to the ambulatory tourist may be minimised by management of the timing of activities and education on the threats in the landscape.

(3)　The capabilities of the animal(s) whose footfalls define the pathway, and the people walking along them, are not equal. This may necessitate deviations around obstacles that can be negotiated by the animal but not the person. Thus trampling impacts from tourists are not fully ameliorated on the pathway. For example, although a large male kangaroo standing at rest may, in this bipedal stance, equate the height of many people, they hop in a more horizontal plane [48] and do not displace equivalent vegetation (Figure 5a). Their capability to hop over objects is also superior to people.

(4)　Defaecation frequently occurs on animal pathways (Figure 5b). It may be random or concentrated at latrines and intersections as a visual and olfactory signpost to communicate use to usually members of the same species. Faecal matter on pathways may be a deterrent to some tourist use.

(5)　The passage of animals along a pathway may over time create erosion and water pooling after rain. The latter may be churned into a muddy patch. For some species like elephants and members of the pig family (Suidae) a muddy patch may be the endpoint or waystation along a pathway. Various research suggests that paths that are eroded and/or muddy may avert tourist use [49].

(6)　Tourists joining an animal pathway may cause an unacceptable disturbance to the species making and using it. For example, they may block the animals' access to water or shelter by displacing them off the pathway. This effect can be ameliorated by a knowledge of the animals' behaviour and ecology, the appropriate timing of tourist activities, and the behaviour of tourists towards the animal (e.g., [50]) in order to minimise disturbance.

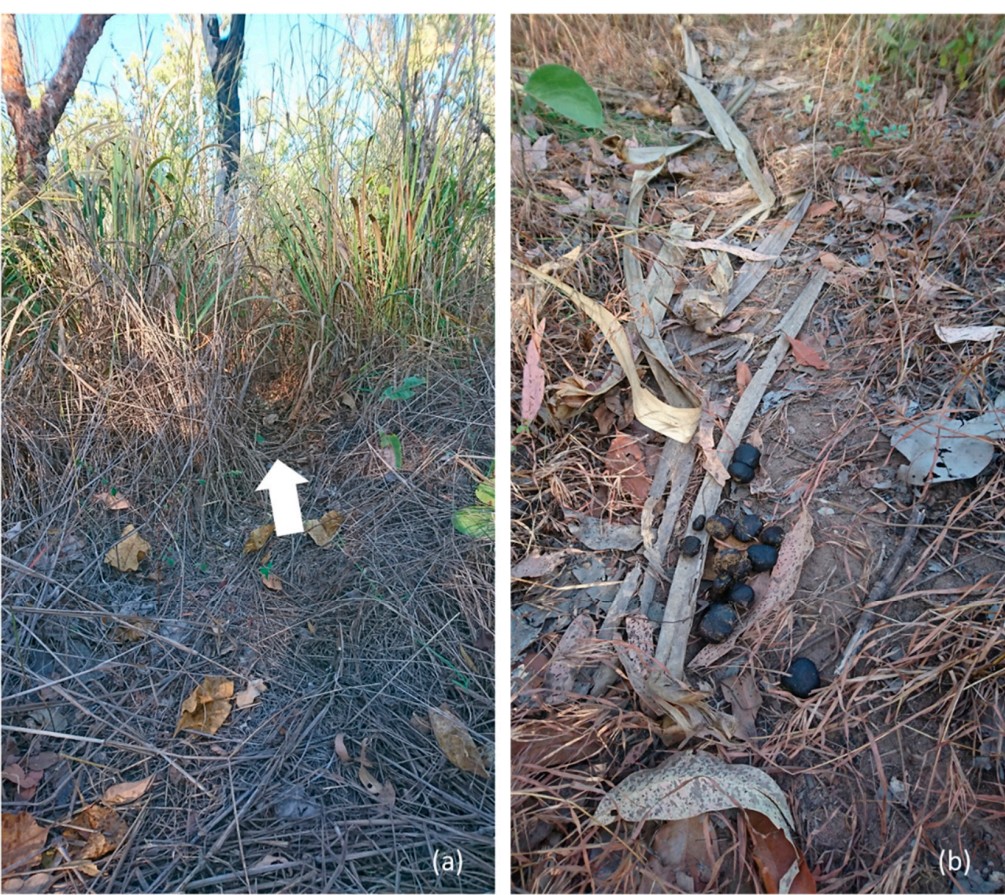

**Figure 5.** (**a**) Animal pathway (arrowed) through tropical savanna vegetation with over-arching vegetation below height of average person, (**b**) faecal deposit from wallaroos (*Osphranter* spp.) deposited on pathway defined by their regular use (Images: D.B. Croft).

## 4. The Trouble with Trampling

Taff and colleagues [51] in this Special Issue, have shown that a large network of informal trails devalues the perception of a landscape for tourist/recreational activity more than trampled/degraded vegetation. The implication was that their survey respondents perceived over-use by people (visitors). This creates a quandary for managers of areas with extensive networks of animal trails if visitors cannot discriminate between informal trails created by the regular use of hikers and those created by the local fauna. An educational response may be necessary to quell perceptions of over-use that may extend beyond people to fauna.

The ecological impacts of formal (and often hardened) and informal trails have been extensively reviewed [52–55]. Pickering and Norman [54] focused on the vegetation beside the trail and concluded more impact from formal hardened trails than informal trails with trail width a qualifying factor. Even so with wet and erodible substrates, hardening has advantages in managing impacts including a reduction in diversions off-trail. In these studies, the origin of informal trails is assumed to be foot-traffic by visitors, and such an assumption may be supported by visitor monitoring using geographic information [56]. Even so some informal trails may have originated as animal trails and have been adopted by hikers, or are primarily created and used by the local fauna. Conradi and colleagues [57] examined 20-year old hiking trails in a calcareous grassland in South Germany in relation to trampling impacts on the composition and diversity of the plant community on and adjacent to the trails. They also looked at animal foraging and seed dispersal as an alternative explanation for the effects they observed. They dismissed the explanatory weight of animal effects, but even so, this is one of few studies that account for both people and the local fauna in trampling impacts.

Barros and Pickering [41] quantified the impacts of hikers and pack animals on informal trails in a high altitude protected area, Aconcagua Provincial Park, in the Andes. They found extensive trampling damage to the vegetation on a landscape scale. The contribution to this informal trail network of "large herds of Guanacos [*Lama guanicoe*]" [58] is not stated. However, the progression of pack animals (mules) caused extensive trampling damage under observed and experimental conditions [59], more so in alpine meadows than steppes where movement off narrow trails was constrained. The indigenous Camelids (Guanacos), with toe bones embedded in a broad cutaneous pad, lack the hard hooves of Equids (mules) and seem to confer no obvious trampling resistance in the vegetation to the introduced Equids. Likewise Pickering and co-authors [60] drew a similar conclusion about the high impacts on vegetation and soils along horse (*Equus caballus*) riding trails in an Australian alpine park. There horses have a short history (around 200 years [61]) compared to indigenous Eastern Grey Kangaroos (2 million years [48]) in the contemporary fauna. They attributed some of this impact to the horses' hard broad hoof compared to the slim foot, comprising a hard nail and cutaneous pad, of the kangaroo. Where horses had a longer, if disjunct, history from 15,000 years ago in North America, horse riding along trails had a lesser impact on soils and vegetation. Some of this resistance is likely conferred by an indigenous North American bovid fauna of hoofed bison, deer and antelope albeit in the Order Artiodactyla not Perrisiodactyla.

The use of pack animals as a proxy to examine impact resistance along tourist trails is not fruitful on most continents. Domestication has a relatively short history focused on a very small number of tractable species [5]. For the horse it dates back to about 3500 B.C.E. on the Eurasian steppes [62]. Even so, controlled experiments on trampling resistance could be conducted with Donkeys (*Equus africanus*) in Africa, Horses in Eurasia and Asian Elephants (*Elephas maximus*) and Water Buffalo (*Bubalus bubalis*) in India and South-East Asia. To date Africa and Asia are grossly under-represented in studies of the ecological impacts of foot or pack-animal traffic along tourism trails [55] so this represents one of many avenues for fruitful research. In this context, it is important to note that animal trails are only recognizable because vegetation and soils are trampled to bare earth (e.g., dry environments) or at the very least vegetation is displaced to the side of the animals' passage (e.g., wet environments). Thus there is a clear impact. Research may focus on whether this is of the same kind as the ambulatory or pack-animal riding tourist, and the degree to which the trails of tourists and local fauna are additive

in their impacts on soils and vegetation (often under protection). Such research should be tempered with the observation, made above, that cropping vegetation may be common in the fauna but is atypical of the tourist's behaviour unless gathering is allowed as part of indigenous culture or for culinary purposes.

Pickering [63] reviewed the resistance of vegetation and ecosystems to trampling by tourists. For ecosystems the ranking by resistance was "subtropical > alpine ~ subalpine ~ arctic ~ temperate > montane" (p. 72) with recognition of significant variation "in resistance within each growth form, climatic zone, and vegetation type". We can ask whether this ranking bears any correlation to the diversity of medium-large herbivorous mammals, species that are likely to create trails through the vegetation. For large herbivores (>100 kg mass), the highest diversity is found either side of the equator on the eastern side of Africa [64] and conforms somewhat to the sub-tropics. This is not replicated on other continents with the other centres of diversity in the Asian steppes extending down through India into South-East Asia. Given Pleistocene extinction of many large herbivores and the rapid lurch towards extinction of the contemporary fauna we are unable to accurately reconstruct the evolutionary pressures of this fauna on trampling resistance. If the body size constraint is relaxed to >2 kg [65] then a high diversity of mammalian herbivores is predicted in the sub-tropics of most continents but there are also pockets of high diversity in temperate zones. The relationships at a global scale between vegetation trampling resistance and mammalian herbivore diversity are, as yet, too imprecise to draw useful conclusions. Pickering [63] rightly recommended site-specific research to uncover relationships to trampling resistance.

To this end, the wet/dry tropics Northern Australia may be a profitable test bed for this research. There are large expanses of tropical savanna woodlands supporting an intact fauna of native herbivores (Macropodidae—kangaroos and wallabies), production zones with introduced cattle (*Bos taurus*) and horses, and feral populations of Equids (Horses, Donkeys), Bovids (Water Buffalo and Banteng *Bos javenicus*) and Suids (Pigs *Sus scrofa*). The vegetation understory may be native or heavily invaded by introduced grasses (typically African), legumes and herbs. There are indigenous cultures of very long tenure that practice some traditions of hunting and gathering with a preference for walking barefoot. The climate offers extremes of defined pluvial (wet) and arid (dry) periods testing the resilience of the vegetation and soils. The evolutionary history of this island continent is relatively uncomplicated.

## 5. Conclusions

There are extensive literatures about trampling impacts on soils and vegetation in animal ecology and tourism research. However, there is little marriage of the two literatures. The methodologies are comparable. Animal ecology has focused on the propagation of grazing and trampling effects from a point source (usually water). Tourism research has focused on trail structure (formal/informal, hardened, wide/narrow) and the propagation of effects (especially weeds) into the hinterland and along the trail. Hominins and animals of equivalent or larger size have been walking across landscapes for millennia and so we would expect some adaptation to this disturbance. Progression of tourists on pack-animals may give some guidance to how one tourist activity replicates animal progression but to date experiments have been undertaken with equids in environments where Perrisiodactyla are not endemic.

We have entered an era of 'overtourism' [66] where small islands of biodiversity aggregate often formerly wide-ranging animal species with a press of humanity coming to view them. Trampling tourists and trampling fauna at high densities have a high potential to degrade an environment and diminish its sustainability as an attraction for tourists and as functioning ecosystems supporting the flora and fauna. Thus destination managers are constrained to limit impacts and so knowledge of whether such impacts are additive in the progression of animals and tourists through the environment is imperative. For trails—formal, informal, animal—such knowledge is compromised by poor definition of what constitutes a trail [55] and quantification as to whether people and other animals walk in each other's footsteps. In other words, do formal trails follow courses traditionally marked out by local

fauna? Do informal trails originate on animal trails given the latter often offer the path of least resistance through the environment and so may attract the deviating tourist? Do the local fauna subsume the trails when vacated by tourists? Canids, like Dingoes (*Canis lupus dingo*), have a propensity to follow man-made roads and paths [67].

If we can get tourists and the larger members of the local fauna on the same path, where the safety of the tourist and the welfare of the animal is at an acceptable risk, then we may mitigate trampling impacts. With the modern armoury of geographically enabled devices from trail cameras to smart/fitness watches there is a rich field of research unfolding. At the very least, trampling impact research in tourism should adopt, the animal trail, as either a 'natural' control along with the traditional (and logical) randomly allocated off-trail measurement path, or as another treatment in assessing landscape effects.

**Funding:** The results presented in Figure 3 derive from a larger study funded by the Australian government through a 'Strategic Partnerships in Research and Training' (SPIRT) grant with the Western Directorate of the NSW National Parks and Wildlife Service.

**Acknowledgments:** I thank my two volunteer assistants, Andrea and Michael, for gathering data contributing to Figure 3 and the staff of Sturt National Park for allowing access to restricted areas of the park. I thank the people of South Africa, Namibia and Botswana for their hospitality to me as a frequent tourist in their national parks.

**Conflicts of Interest:** The author declares no conflict of interest.

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
