# Peer review of "Walking in Each Other’s Footsteps: Do Animal Trail Makers Confer Resilience against Trampling Tourists?"

_environments, doi:10.3390/environments6070083_

Reviewer 1 Report

 I thought that this is a potentially fascinating piece of research which, as noted, is understudied. I have only some small suggestions to make

a) a little more comprehensive overview of the trampling problem by tourists would be welcome

b) in the North American parks many of the first walking trails were laid out on wildlife trails because of ease of construction - i have seen this referred to several times in park histories but am struggling to locate a reference!!!

Author Response

I thank the reviewer for the interest expressed in the topic of the manuscript. 

The effects of human trampling have been extensively reviewed and these reviews are referred to in section 4. Even so to accommodate the request for more information on human trampling I have added some additional material (e.g. lines 294-298).

Reviewer 2 Report

Manuscript “Walking in each other’s footsteps: do animal trail makers confer resilience against trampling tourists?” presents a literature overview of the tourists trampling in the natural environment. Both well-trapped areas and informal routes devalue the perception of the landscape.

The article is a valuable review material. Properly selected literature guarantees the understanding of the discussed problem.

Comments for authors

There are no major remarks to the language.

Subject

It corresponds to the content.

Keywords

The keywords chosen correctly.

Abstract

The summary reflects the content of the article.

1. Introduction

Correct introduction to the topic. Clearly specified aims for the review.

2. Trampling impacts of the hooves and hard landings

The chapter based on the research conducted in Africa and Australia. A question arises, why there are no references to other continents? I suggest supplementing or changing the subject – the limitation to two presented continents.

No data about the author of the photos (1, 2).

3. Tourist and wildlife in other’s footsteps

The chapter also applies to Africa and Australia. I suggest supplementing or changing the subject – limitation to two presented continents.

 No data about the author of the photos (3, 4, 5).

4. The trouble with trampling

The chapter includes content about South America and Asia. I suggest supplementing the other chapters or removing the part concerning Asia and South America (with the change of the subject at the same time). The review will be more consistent in terms of spatial research presented.

5. Conclusions

The conclusions are consistent with the aims of the work and the whole review. They discuss the deficiencies in the subject literature and indicate future research directions. They contain the general recommendations for mitigating the effects of environmental trampling by tourists.

Conclusion from the review - the manuscript requires minor additions and changes.

Author Response

All images were taken by the author. The appropriate attribution is made for each figure and I thank the reviewer for reminding me to do so. The reviewer rightly recognises some geographic bias towards Africa and Australia in sections 2 and 3. In respect to Africa this somewhat counters the bias in the human trampling literature against Africa, as stated in section 4, but where Australia is well-represented.

I have added new material to counter my bias and made explicit the location of examples I have given as follows:

Lines 72-78 – cited literature from USA, Argentina, Iran and Mongolia.

Line 126-132. Added an additional site as an exemplar albeit also from Africa.

Lines 163-170. Added a paragraph explaining geographic bias and how the findings may be generalised on a global scale and between animal and human trampling. This is based on insights from research in the USA cited in lines 72-78.

Lines 176-179. Added example from tundra in Sweden and explained paucity of information regardless of geography. Added lines 180-183 to explain current focus on animal trails which is monitoring with trail cameras (camera traps). I could have included extensive use of trail cameras in recreational hunting in North America but feel that is a bit off-topic.

Line 208: Added location of research which was the USA.

Line 291-298. Added a European example.